# Homology Consistency Constrained Efficient Tuning for Vision-Language Models

**Huatian Zhang, Lei Zhang, Yongdong Zhang, Zhendong Mao**[*]
University of Science and Technology of China
huatianzhang@mail.ustc.edu.cn, {leizh23,zhyd73,zdmao}@ustc.edu.cn

## Abstract

Efficient transfer learning has shown remarkable performance in tuning large-scale vision-language models (VLMs) toward downstream tasks with limited data resources. The key challenge of efficient transfer lies in adjusting image-text alignment to be task-specific while preserving pre-trained general knowledge. However, existing methods adjust image-text alignment merely on a set of observed samples, *e.g.*, data set and external knowledge base, which cannot guarantee to keep the correspondence of general concepts between image and text latent manifolds without being disrupted and thereby a weak generalization of the adjusted alignment. In this work, we propose a Homology Consistency (HC) constraint for efficient transfer on VLMs, which explicitly constrains the correspondence of image and text latent manifolds through structural equivalence based on persistent homology in downstream tuning. Specifically, we build simplicial complex on the top of data to mimic the topology of latent manifolds, then track the persistence of the homology classes of topological features across multiple scales, and guide the directions of persistence tracks in image and text manifolds to coincide each other, with a deviating perturbation additionally. For practical application, we tailor the implementation of our proposed HC constraint for two main paradigms of adapter tuning. Extensive experiments on few-shot learning over 11 datasets and domain generalization demonstrate the effectiveness and robustness of our method.

## 1 Introduction

Large-scale vision-language models (VLMs) such as CLIP [1] and ALIGN [2] trained on web-scale data have learned broad visual concepts and demonstrated promising generalization capability on a wide range of downstream tasks, such as classification [3, 4], detection [5, 6] and segmentation [7, 8]. In transferring to downstream tasks with limited data resources, the conventional full fine-tuning on VLMs often forgets the general knowledge learned in pre-training and falls into overfitting. To mitigate this, how to efficiently transfer the knowledge from pre-trained VLMs to downstream tasks in a low-data regime has been intensively studied.

To possess both task-specific knowledge exploration and general knowledge preservation, efficient transfer learning proposes to adapt VLMs to fit downstream tasks by tuning a few parameters, which mainly exists in two paradigms: prompt tuning and adapter tuning. Prompt tuning methods adapt VLMs toward downstream tasks by introducing learnable prompts on the input side. Topics in this branch include the configuration of learnable prompts [9–13] and injecting semantic priors, such as external knowledge [14, 15], category distribution [16] or visual diversity [17, 18], in tuning. As a promising alternative, adapter tuning inserts a learnable lightweight adapter into the frozen pre-trained VLMs on the output side and the insertion manner allows modifying flexibly. By residual blending, CLIP-Adapter [19] tuned the image or text embeddings by appending a learnable bottleneck layer

---

[*]Corresponding author.

38th Conference on Neural Information Processing Systems (NeurIPS 2024).

to the frozen encoder. TaskRes [20] added learnable parameters as prior-independent residuals to text embeddings. Another line is based on key-value cache model. Tip-Adapter [21] proposed to construct an adapter with training images as keys and one-hot label encodings as values for image recognition. Some work further enhanced the cache model by discriminative prior refinement [22] or knowledge augmentation [23]. For efficient transfer, existing methods mainly focus on how to configure tunable parameters or the leverage of external prior knowledge in tuning.

The essence of task-specific tuning on VLMs is to adjust the semantic alignment of image and text latent manifolds to fit downstream tasks. In this view, efficient transfer learning aims to establish new semantic alignments while keeping the correspondence of pre-trained general concepts between image and text latent manifolds from being corrupted. This correspondence stems from the equivalence of semantics between the two manifolds. However, existing methods adjust image-text alignment toward downstream tasks on a set of observed samples from the latent manifolds. The discrete samples are incapable of adequately capturing the underlying structure of manifolds. The lack of perspective on latent manifolds in alignment adjusting risks devolving the desired manifold equivalence into localized closeness on the observed data, particularly in a low-data regime, which causes the unguaranteed generalization of adjusted alignment beyond the data samples.

To this end, we propose to explicitly constrain the equivalence of image and text latent manifolds in transferring VLMs toward downstream tasks. We study the structure of the image and text latent manifolds from the lens of topological data analysis [24–26]. Topology encodes the connectivity of a space to describe its underlying structure, and the preservation of topology between spaces is fundamental for their structural equivalence. Topological data analysis employs homology groups to quantify the topological features of manifold structure, such as connected components, loops, voids, and higher-dimensional holes, as homology classes and tracks the survival of the topological features across multiple scales via persistent homology [26] to capture their size and position, which summaries the global shape of manifold. These topology insights provide an avenue to achieve a structural equivalence of image and text latent manifolds.

In this work, we propose a Homology Consistency (HC) constraint for efficient transfer on VLMs, which constrains the structural equivalence of image and text latent manifolds based on persistent homology in downstream tuning. Given image-text data samples, we construct simplicial complex on the top of data to mimic the topological structure of latent manifolds, and induce a nested sequence of subcomplexes called filtration. Through the filtration, we capture the persistence of homology classes from their appearance to non-existence and guide the homology persistences of image and text manifolds to be consistent. Specifically, we locate the births and deaths of homology classes in latent manifolds and track these homology persistences, then guide the directions of persistence tracks in image and text manifolds to coincide each other so as to achieve a homology-level structural equivalence. Additionally, we apply a deviating perturbation to persistence-related text samples to encourage their respective semantically related images to be distributed uniformly relative to them in embedding, in order to enhance the generalization of the track coincidence in adjusting image-text alignment. Further, we tailor the implementation of the proposed HC constraint for the main residual blending and key-value cache based paradigms of adapter tuning. Extensive experiments on few-shot learning over 11 benchmarks and domain generalization demonstrate the effectiveness and robustness of HC constraint in efficient transfer learning on VLMs.

Our main contributions are summarized as follows:

- We propose to explicitly constrain the structural equivalence of image and text latent manifolds in efficient transfer on VLMs, to improve the generalization of downstream image-text alignment adjusting beyond data samples in a low-data regime.

- We propose a theoretically well-founded homology consistency (HC) constraint based on persistent homology for efficient transfer on VLMs. We coincide the persistences of homology classes of topological features between image and text manifolds, and apply a deviating perturbation to generalize the persistence coincidence to unseen data.

- We tailor the implementation of the proposed HC constraint for the two main paradigms of adapter tuning respectively, showing the transferability of our method.

- We evaluate the proposed HC constraint on few-shot classification over 11 popular benchmarks. The extensive experiments demonstrate that HC constraint can boost the performance of baselines significantly and achieve state-of-the-art.

## 2 Related Work

**Efficient Transfer Learning.** To better transfer VLMs to downstream tasks especially with limited target domain data, a lot of research on efficient transfer learning has been done, which mainly exists in two types, prompt tuning and adapter tuning. Advanced than manual prompt that demands domain expertise to develop suitable format, prompt tuning methods design learnable prompts to adapt VLMs on downstream data. As a pioneer work, CoOp [9] for the first time composed prompts by concatenating text category embedding and learnable context vectors. The learnable prompts can be configured in text input [9, 10], image input [11] or jointly both [12, 13]. A line of work focuses on injecting semantic priors, such as category-related external knowledge [14, 15], category embedding distribution [16] and the diversity of visual concepts [17, 18], in prompt tuning. In another branch, adapter tuning methods insert a learnable lightweight adapter module into the frozen pre-trained VLMs and show excellent performance. The adapter architecture and insertion manner allow for flexible modifications. There are two main paradigms: residual blending and key-value cache based. By residual blending, CLIP-Adapter [19] appended learnable bottleneck layer to frozen encoder to tune embeddings. TaskRes [20] added a set of prior-independent parameters to frozen text category embeddings to obtain an image classifier. GraphAdapter [27] proposed to learn downstream knowledge with inter-class relationship of image and text samples. Based on the key-value cache model, Tip-Adapter [21] constructed an adapter with training images and label encodings as key-value to recognize query images. APE [22] refined the cache model by visual discrimination priors. CaFo [23] cascaded diverse external knowledge from DINO [28], DALL-E [29], and GPT-3 [30] to assist recognition. However, although existing methods have attained remarkable achievements in VLMs transfer, they adjust the semantic alignment [31–33] of image and text latent manifolds toward downstream tasks merely on observed samples, *e.g.*, data sets and external knowledge bases, and lack insight into underlying manifold structure, which may cause unguaranteed generalization beyond the data samples. In this work, we propose to explicitly constrain the structural equivalence of image and text latent manifolds in downstream tuning to facilitate the transfer of VLMs.

**Topological Data Analysis in Machine Learning.** The area of topological data analysis [26] infers the topological structure of data spaces using algebraic tools such as persistent homology, and has been applied in many fields of machine learning, *e.g.*, image segmentation [34–37], graph machine learning [38–40], molecular representation [41, 42], point cloud analysis [43, 44], etc. For instance, in image segmentation, [35] proposed to drive the segmentations to contain the specified topological features without requiring ground-truth labels. [36] used discrete Morse theory and persistent homology to learn the structural representation of images for fine-scale structure segmentation. In graph machine learning, [39] integrated vertex- and edge-level topological features into message-passing graph neural networks to boost their expressive power. In point cloud analysis, [44] developed a learnable filtration on point clouds to obtain adaptive topological features for given tasks. Besides, [45] preserved the topological structures of input space into latent space of autoencoders by aligning topologically relevant distances. [46] applied representation topology divergence [47] in dimensionality reduction to force closeness on topological structures. How to analyse the structure of data spaces for semantic alignment in vision-language tasks through topology remains under-explored in the literature. In this work, we characterize and align the structure of the image and text latent manifolds by means of persistent homology in efficient transfer learning on VLMs.

## 3 Methodology

### 3.1 Preliminaries

#### 3.1.1 Contrastive Language-Image Pre-training (CLIP)

As a representative VLM, CLIP [1] is trained on massive image-text pairs and shows promising zero-shot performance on downstream tasks. CLIP adopts two separate encoders to embed images and texts into latent manifolds, and aligns the bi-modal embeddings by contrastive learning that forces paired image and text closer and unpaired ones away. For the transfer to image classification with $N$ classes, CLIP obtains the class embedding $w_i$ by feeding the prompt templates, *e.g.*, "A photo of a [CLASS]", filled with class name $c_i$ into text encoder, and the probability that an image $x$ belongs to category $c_i$ can be formulated as $p(y = c_i|x) = \exp\left(x^\top w_i/\tau\right) / \sum_{j=1}^{N} \exp\left(x^\top w_j/\tau\right)$, where the embeddings are $l_2$-normalized and $\tau$ denotes a temperature hyper-parameter.

### 3.1.2 Persistent Homology

Discrete data points contain only observations from the latent space in which they reside and do not have interesting topology. To peek at the topological structure of latent space, we mimic its connectivity by constructing a simplicial complex with the data points as vertices. As a basic element, simplex $\sigma$ with dimension $p$ is the convex hull of a set of $p + 1$ affinely independent points $(x_0, \ldots, x_p)$. Given a finite data point set $X$ in metric space $(M, d)$ and a threshold $a > 0$, the commonly used Vietoris-Rips (Rips in short) complex is defined as:

$$K_a(X) = \{\sigma \subset X \mid d(x_i, x_j) < a, \forall x_i, x_j \in \sigma\}, \tag{1}$$

which is fully determined by the pairwise distances of $X$. The dimension of $K_a$ is the maximum dimension of any simplex within it. The formal sums of $p$-simplices added with $\mathbb{Z}_2$-additions form a chain group $\mathrm{C}_p(K_a)$ ($\mathrm{C}_p$ for brevity). Then, define a boundary operator $\partial_p$ on $p$-simplex $\sigma$ as a map that sends $\sigma$ to the $(p-1)$-chain consisting of $\sigma$'s $(p-1)$-faces referred as $\sigma$'s boundary. Applying $\partial_p$ to the chain groups can obtain a sequence of homomorphisms: $\mathrm{C}_k \xrightarrow{\partial_k} \mathrm{C}_{k-1} \cdots \mathrm{C}_1 \xrightarrow{\partial_1} \mathrm{C}_0$. All $p$-chains whose boundaries are empty form a cycle group $\mathrm{Z}_p$, which is the kernel of $\partial_p$. The image of boundary operator $\partial_{p+1}$ on $\mathrm{C}_{p+1}$ forms a boundary group $\mathrm{B}_p$. Further, taking the quotient of the $\mathrm{Z}_p$ with $\mathrm{B}_p$, the $p$-th homology group $\mathrm{H}_p = \mathrm{Z}_p/\mathrm{B}_p$ classifies the $p$-cycles in $\mathrm{Z}_p$ by collecting those cycles that differ by a boundary into the same homology class. In a topological view, the rank of homology group $\mathrm{H}_p$ captures the number of $p$-dimensional holes in space.

Persistent homology offers a way to compute the quantified summary of topological structures of the latent space from sampled data. For the data point set $X$ in space $(M, d)$, we define a function $f : M \to \mathbb{R}, f((x_0, \ldots, x_p)) = \max_{i,j \in \{0,\ldots,p\}} f((x_i, x_j))$ on simplices. Then, given a sequence of thresholds $a_1 \leq a_2 \leq \ldots, \leq a_n$, the growing sublevel sets $f^{-1}(-\infty, a]$ at these values give rise to a nested sequence of subcomplexes, $K_{a_1} \subseteq K_{a_2} \subseteq \cdots \subseteq K_{a_n}$, called a filtration $\mathcal{F}$, as shown in Fig. 1. The inclusions in $\mathcal{F}$ induce:

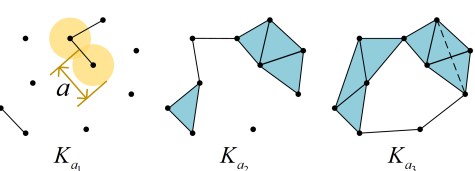

$$K_{a_1} \qquad K_{a_2} \qquad K_{a_3}$$

Figure 1: The sublevel set filtration on a nested family of Rips complexes.

$$0 = \mathrm{H}_p(K_0) \to \cdots \to \mathrm{H}_p(K_i) \xrightarrow{h_p^{i,j}} \mathrm{H}_p(K_j) \to \cdots \to \mathrm{H}_p(K_n) = \mathrm{H}_p(K), \tag{2}$$

where the images of the homomorphisms $h_p^{i,j}$ are persistent homology groups $\mathrm{H}_p^{i,j}$. A non-trivial homology class $\epsilon \in \mathrm{H}_p(K_a)$ is born at $K_i$, if $\epsilon \in \mathrm{H}_p^{i,a}$ but $\epsilon \notin \mathrm{H}_p^{i-1,a}$. Likewise, the homology class $\epsilon$ dies entering $K_j$, if $\epsilon \in \mathrm{H}_p^{a,j-1}$ but $\epsilon \notin \mathrm{H}_p^{a,j}$. The persistence of a homology class is the lifespan from its birth to death. See Appendix A for more details.

### 3.2 Homology Consistency

Given a set of pre-trained image embeddings $X$ and text embeddings $T$, we construct a Rips complex $K_{a_M}(X)$ with the maximum pairwise distance $a_M$ of $X$ and further derive a sublevel set filtration, $\mathcal{F}(K)$, as the nested family of subcomplexes $K_{a_0} \subseteq K_{a_1} \subseteq \cdots \subseteq K_{a_M}$ at the increasing scale sequence of pairwise distances $\{a_i\}_{i=0}^M$ where $a_0 = 0$. Then we arrive at $p$-th persistent homology groups that capture the survival of the homology classes of $p$-dimensional topological features (*e.g.*, 0-dimension: connected components, 1-dimension: loops, 2-dimension: voids, etc.) and pair the birth and death times of $p$-th homology classes, following [45]. Since the edge skeleton of Rips complex fully determines all of its simplices,

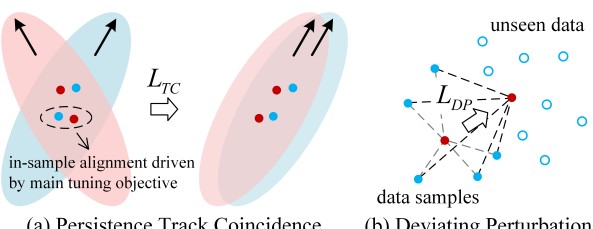

(a) Persistence Track Coincidence     (b) Deviating Perturbation

Figure 2: Schematic illustration of our proposed HC constraint. (a) TC guides the directions of the persistence tracks to coincide each other to establish the alignment of underlying structures beyond the observed samples. (b) DP encourages samples to be uniformly distributed.

under the assumption that pairwise distances of $X$ are unique, every birth-death time pair can be mapped back to the simplices that respectively created and destroyed the unique corresponding homology class. The schematic illustration of HC constraint is shown in Fig. 2.

**Persistence Track Coincidence.** Let $\mu$ be a $p$-th homology class that is born at $a_b$ and dies at $a_d$, then the birth simplex $\beta$ forms $\mu$ in $K_{a_b}$ and the death simplex $\delta$ causes $\mu$ to disappear entering $K_{a_d}$. For example, if $\mu$ were to be 0-dimensional, it emerges at $a_0$, and $\delta$ is the edge that joins it with some other point; if $\mu$ were to be 1-dimensional, then $\beta$ is the edge that forms the loop corresponding to $\mu$ in $K_{a_b}$, and $\delta$ is the triangle that incurs the loop to be contractible in $K_{a_d}$. Since the Rips complex is pairwise distance based, on every $p$-simplex we have:

$$f\left((x_0, \ldots, x_p)\right) = \max_{i,j \in \{0,\ldots,p\}} f\left((x_i, x_j)\right). \tag{3}$$

For the birth simplex $\beta$ and death simplex $\delta$, this means they will not be established until their longest edge $(x_{i(\beta)}, x_{j(\beta)})_{\arg\max f(\beta)}$ and $(x_{i(\delta)}, x_{j(\delta)})_{\arg\max f(\delta)}$ appear at time $a_b$ and $a_d$, which we refer to as birth edge and death edge, respectively. It can be said that the simplex $\beta$ is completed by the birth edge and $\delta$ by the death edge. Such edges mark the creation and destruction of the homology class $\mu$ in the image latent manifold. Defining the direction of the edges as from $i$ to $j$ such that $i < j$ in the given vertex sequence of simplices, we track the persistence direction of $p$-th homology class $\mu$ from its birth edge to death edge as:

$$\mathrm{tra}_p(\mu, X) = (x_{i(\delta)}, x_{j(\delta)})_{\arg\max f(\delta)} - (x_{i(\beta)}, x_{j(\beta)})_{\arg\max f(\beta)}. \tag{4}$$

Then we obtain the persistence track of $\mu$ in text latent manifold, $\mathrm{tra}_p(\mu, T)$, through determining its birth and death edges directly by taking the corresponding texts of the end-points of $\mu$'s birth and death edges in image manifold accordingly. Further, for aligning the structure of image and text latent manifolds, we guide the track coincidence (TC) of $p$-th homology classes between the two by:

$$L_{TC}(\Gamma_p, X, T) = 1 - \frac{1}{|\Gamma_p|} \sum_{\mu \in \Gamma_p} \varphi(\mathrm{tra}_p(\mu, X), \mathrm{tra}_p(\mu, T)), \tag{5}$$

where $\Gamma_p$ is the set of $p$-th homology classes in latent manifolds and $\varphi$ is cosine similarity.

**Deviating Perturbation.** In a low-data regime where samples are not sufficient to fully characterize the topology of the latent manifold of interset, the sight of persistence track coinciding is confined on the given limited samples, which hinders the generalization of structural equivalence guided by track coincidence. For all end-point images of birth and death edges of $p$-th homology classes in $\Gamma_p$ and their corresponding texts, we drive every text to deviate from its semantically related images in embedding (what we consider here is that multiple images are related to the same category text in classification) without breaking the track coincidence, so as to encourage the text-related images beyond samples in latent manifold to be uniformly distributed around the text.

Specifically, since the persistence tracks of the 0-th homology classes are exactly the death edges (birth edges are 0), their end-point samples are available from the given dataset. We quantify the degree of deviation as the similarity between the orientations of the text's embedding relative to the embeddings of its related images. Then, for 0-th homology classes in $\Gamma_0$, we enlarge the deviation between end-point texts and their respective semantically related images by reducing the variation in their relative orientations, by applying a deviating perturbation (DP) as:

$$L_{DP}(\Gamma_0, X, T) = \frac{1}{|X_{\Gamma_0}|} \sum_{x_i \in X_{\Gamma_0}} (1 - \frac{1}{|X_i'|} \sum_{x_i' \in X_i'} \varphi(x_i - (x_i)_T, x_i' - (x_i)_T)), \tag{6}$$

where $X_{\Gamma_0}$ denotes the set of all end-point images of 0-th homology classes in $\Gamma_0$, $(x_i)_T$ denotes $x_i$'s corresponding text, $X_i' \in X_{\Gamma_0}$ denotes a set of images of the same category that are semantically related to $x_i$ and $\varphi$ is cosine similarity. The deviating perturbation can benefit the structural equivalence to get rid of a biased reconstruction of latent topology caused by insufficient sampling.

**Homology Consistency.** To constrain the structural equivalence of image and text latent manifolds, we coincide persistence tracks of homology classes along with the deviating perturbation by:

$$L_{HC}(\Gamma, X, T) = \sum_{p=0}^{n} L_{TC}(\Gamma_p, X, T) + \lambda L_{DP}(\Gamma_0, X, T), \tag{7}$$

where $\lambda$ is the hyper-parameter controlling the perturbation strength. We mainly focus on the 0-th homology classes ($p = 0$), since higher-dimensional classes significantly increase computational cost but bring almost no additional performance benefits in VLMs tuning in practice.

### 3.3 Implementation

To examine the structural equivalence constrained by our proposed homology consistency in the efficient transfer of VLMs and show its transferability, we tailor the implementation of homology consistency constraint to the two main paradigms of adapter tuning, residual blending based and key-value cache model based, respectively.

**Residual blending based** tuning methods construct an adapter to produce learnable residuals and blend them with the pre-trained features. As a representative, TaskRes [20] adds prior-independent parameters $x$ as a residual to the pre-trained text embeddings $t$ to form a learnable image classifier $t' = t + \alpha x$, where $\alpha$ is a scaling factor, and updates the classifier by cross-entropy loss $L_{CE}$.

In this paradigm, given frozen pre-trained image embeddings $X$ and tunable text embeddings $T$, our method can naturally construct a filtration on $X$ to capture the image persistence tracks and further obtain their text tunable counterparts, and then employ the persistence track coincidence with deviating perturbation through $L_{HC}(\Gamma, X, T)$ in downstream tuning together with $L_{CE}$.

**Key-value cache based** tuning methods build adapters via a key-value cache model with pre-trained embeddings of all training images as keys and one-hot encodings of corresponding labels as values. The image keys can be unfreezed as learnable parameters. As a representative, to recognize an image $x$, Tip-Adapter-F [21] first measures its affinity weights with the cached keys $F_{\text{train}}$ by $A = \exp\left(-\beta\left(1 - xF_{\text{train}}^T\right)\right)$, then aggregates the cached values $L_{\text{train}}$ with weights $A$ as a prediction $AL_{\text{train}}$, and further combines $AL_{\text{train}}$ with the similarity between image $x$ and pre-trained text category embeddings $W_c$ as final classification logit $\alpha AL_{\text{train}} + xW_c^T/\tau_{\text{CLIP}}$ in $L_{CE}$.

In this paradigm, the cache-based adapter represents the categories as one-hot label encodings $L_{\text{train}}$, that is, the $L_{\text{train}}$ is the only textual representation of categories in this adapter. To be comparable to label encodings, we regard the affinity weights $A$ of an image as a sparse visual representation of this image. For implementing homology consistency, we first capture the birth and death edges of image homology classes based on pre-trained image embeddings, then replace end-point images of the edges with their corresponding affinity weights $A$ or label encoding $L_{\text{train}}$ and follow Eq. 4 to construct $\text{tra}_p(\mu, A)$ and $\text{tra}_p(\mu, L_{\text{train}})$ analogically. We implement HC as $L_{HC}(\Gamma, A, L_{\text{train}})$ by taking $\text{tra}_p(\mu, A)$ and $\text{tra}_p(\mu, L_{\text{train}})$ as proxies for original image and text tracks.

**Optimization.** Efficient transfer learning commonly adopts the cross-entropy loss $L_{CE}$ between labels and the predicted class probability to tune learnable parameters, *e.g.*, $\theta$, on downstream classification tasks. We have two ways to integrate the gradient from our proposed HC constraint into parameter tuning. (1) We alter the cross-entropy gradient $\nabla_\theta L_{CE}$ toward the direction of HC by adding the gradient of HC constraint to $\nabla_\theta L_{CE}$ with a constant factor $\eta$ by: $\nabla_\theta L = \nabla_\theta L_{CE} + \eta\nabla_\theta L_{HC}$. (2) For Tip-Adapter-F, due to its slightly complex hyper-parameter configuration, fixing a constant factor is difficult to control the contribution of each gradient. We adaptively keep the gradients at the same order of magnitude through scaling the gradient of homology consistency constraint based on gradient norm by $\nabla_\theta L = \nabla_\theta L_{CE} + \omega\frac{\|\nabla_\theta L_{CE}\|_2}{\|\nabla_\theta L_{HC}\|_2}\nabla_\theta L_{HC}$ in optimization.

## 4 Experiments

### 4.1 Experimental Settings

**Datasets.** Following previous efficient transfer learning works, we conduct the few-shot learning evaluation on 11 benchmark datasets including Caltech101 [48], DTD [49], EuroSAT [50], FGVCAircraft [51], Flowers102 [52], Food101 [53], ImageNet [54], OxfordPets [55], StanfordCars [56], SUN397 [57] and UCF101 [58]. These datasets cover a wide range of visual recognition on generic objects, fine-grained categories, scenes, actions, etc. We sample 1, 2, 4, 8 and 16 shots per class, respectively, for model training and evaluate on full test sets. In addition, we evaluate the domain generalization performance of our method with the ImageNet [54] as source and its variants ImageNetV2 [59], ImageNet-Sketch [60], ImageNet-A [61] and ImageNet-R [62] as targets.

**Implementation Details.** In addition to HC, following TaskRes, we augment the original pre-trained text features into ones tuned on downstream tasks in HC$^*$. In implementing HC / HC$^*$ on TaskRes (HC-TR / HC$^*$-TR), we set the values of $\eta$ and $\lambda$ on different datasets according to the procedure of factor determination in ablation studies. We set the scaling factor $\alpha$ to 1 for all datasets. The training batch size is 256. We employ the Adam optimizer with an initial learning rate of $1e^{-4}$

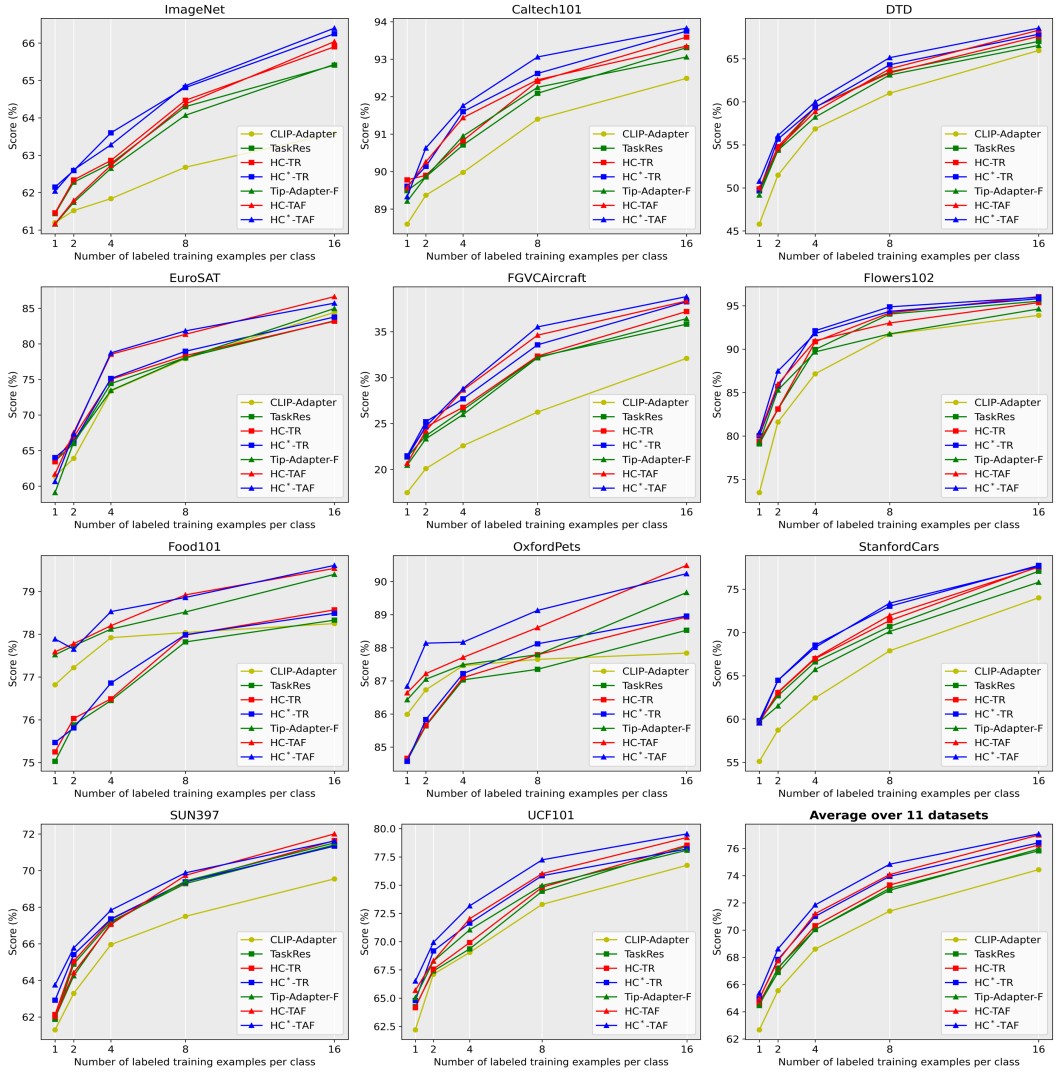

Figure 3: The performance comparison of baselines and our proposed HC and HC$^*$ on few-shot learning, including 1-/2-/4-/8-/16-shots on 11 benchmark datasets and the average accuracy. The full numerical results are provided in the Appendix B.

on ImageNet and $1e^{-3}$ on others, and the learning rates decay with cosine learning rate schedule following TaskRes. In implementing HC / HC$^*$ on Tip-Adapter-F (HC-TAF / HC$^*$-TAF), $\omega$ and $\lambda$ are also set according to the factor determination procedure in ablation studies. The $\lambda$ here is significantly larger than the above because the magnitude difference between the $\nabla_\theta L_{TC}$ and $\nabla_\theta L_{DP}$ here is relatively large. The training batch size is $256$. Following Tip-Adapter-F, we employ the AdamW optimizer with a cosine annealing scheduler. We set initial learning rate as $1e^{-3}$. All experiments are conducted on a single NVIDIA A40 GPU. Note that the experiments on baseline and with the HC / HC$^*$ are conducted in the same setting for fair comparison. The code is publicly available [2].

## 4.2 Performance Analysis

**Few-shot Learning.** We validate the effectiveness of our proposed HC constraint on 11 benchmark datasets using representative adapter tuning methods, the residual blending based TaskRes and key-value cache based Tip-Adapter-F, as baselines. The experimental results are shown in Fig. 3, from which it can be observed that our method brings performance improvements for 1 to 16 shots

---

[2]https://github.com/htzhang-code/HC

consistently. Taking the 16 shots average accuracy as an example, HC-TR exceeds TaskRes by $0.40\%$ and HC-TAF exceeds Tip-Adapter-F by $1.04\%$. At 16-shot, HC-TAF outperforms Tip-Adapter-F by $0.61\%$ on ImageNet. For the challenging fine-grained classification dataset FGVCAircraft, the proposed HC constraint gains $1.38\%$ and $1.92\%$ on TaskRes and Tip-Adapter-F, respectively. The further improvements of HC$^*$-TR and HC$^*$-TAF yielded by pre-tuned text classifier suggest that the HC-equipped tuning can still significantly benefit from representation enhancement.

The performance comparison of our HC constraint with the state-of-the-art adapter tuning methods on ImageNet is shown in Tab. 1. APE-T performs markedly best at 1 and 2 shots, whereas the models constrained by homology consistency, *e.g.*, HC$^*$-TAF, exceed it and other state-of-the-arts on the 8-/16-shot setting in more sample cases. This arises from the fact that the efficacy of homology consistency in vision-language aligning depends primarily on the modeling capability of the simplicial complex built on training data towards the topological structure of latent space. The

Table 1: The performance comparison of our methods with the state-of-the-art methods on ImageNet.

| Method | 1-shot | 2-shot | 4-shot | 8-shot | 16-shot |
|---|---|---|---|---|---|
| Zero-shot CLIP [1] | 58.18 | 58.18 | 58.18 | 58.18 | 58.18 |
| CLIP-Adapter [19] | 61.20 | 61.52 | 61.84 | 62.68 | 63.59 |
| TaskRes [20] | 61.44 | 62.28 | 62.78 | 64.30 | 65.41 |
| Tip-Adapter-F [21] | 61.16 | 61.74 | 62.65 | 64.07 | 65.43 |
| GraphAdapter [27] | 61.50 | 62.32 | 63.12 | 64.23 | 65.70 |
| APE-T [22] | **62.51** | **63.25** | **63.66** | 64.80 | 66.07 |
| HC-TR (Ours) | 61.46 | 62.34 | 62.86 | 64.47 | 65.90 |
| HC-TAF (Ours) | 61.17 | 61.79 | 62.73 | 64.37 | 66.04 |
| HC$^*$-TR (Ours) | 62.15 | 62.59 | 63.60 | 64.81 | 66.25 |
| HC$^*$-TAF (Ours) | 62.04 | 62.61 | 63.28 | **64.86** | **66.40** |

denser the data samples, the more sufficiently they summarize the structure, and the more effective the homology consistency constraint is. It is worth noting that HC boosts baselines to achieve state-of-the-art without introducing any additional training parameters. We respectively take TaskRes and Tip-Adapter-F as representative methods of residual blending based (*e.g.*, CLIP-Adapter, TaskRes, GraphAdapter) and key-value cache based (*e.g.*, Tip-Adapter-F, APE-T) adapter tuning for applying HC constraint. The HC constraint is verified to be effective on these two baselines by extensive experiments, and can theoretically be extended to other VLMs efficient transfer learning methods.

Table 2: The performance comparison on domain generalization over four CLIP visual backbones. All methods are trained on the ImageNet in 16-shot setting and evaluated on the domain-shifted datasets, ImageNet-V2, -Sketch, -A, and -R.

| Method | Backbone | Source | Target | | | | |
|---|---|---|---|---|---|---|---|
| | | ImageNet | -V2 | -Sketch | -A | -R | Average |
| Zero-shot CLIP [1] | | 58.18 | 51.34 | 33.32 | 21.65 | 56.00 | 40.58 |
| Linear Probe CLIP [1] | | 55.87 | 45.97 | 19.07 | 12.74 | 28.16 | 28.16 |
| TaskRes [20] | ResNet-50 | 65.41 | 56.84 | 35.54 | 21.68 | 59.96 | 43.51 |
| Tip-Adapter-F [21] | | 65.43 | 57.20 | 35.99 | **23.52** | 60.45 | 44.29 |
| HC-TR (Ours) | | 65.90 | 56.97 | 35.36 | 21.20 | 59.57 | 43.28 |
| HC-TAF (Ours) | | **66.04** | **57.44** | **36.17** | 23.49 | **60.52** | **44.41** |
| Zero-shot CLIP [1] | | 61.62 | 54.81 | 38.71 | 28.05 | 64.38 | 46.49 |
| Linear Probe CLIP [1] | | 59.75 | 50.05 | 26.80 | 19.44 | 47.19 | 35.87 |
| TaskRes [20] | ResNet-101 | 68.26 | 59.94 | 41.30 | 28.91 | 67.36 | 49.38 |
| Tip-Adapter-F [21] | | 68.47 | 59.69 | 41.63 | 30.05 | 68.04 | 49.85 |
| HC-TR (Ours) | | 68.62 | 59.66 | 41.12 | 29.07 | 66.97 | 49.21 |
| HC-TAF (Ours) | | **68.80** | **60.30** | **41.76** | **30.08** | **68.15** | **50.07** |
| Zero-shot CLIP [1] | | 62.05 | 54.79 | 40.82 | 29.57 | 65.99 | 47.79 |
| Linear Probe CLIP [1] | | 59.58 | 49.73 | 28.06 | 19.67 | 47.20 | 36.17 |
| TaskRes [20] | ViT-B/32 | 68.45 | 59.54 | 42.09 | 30.60 | 68.80 | 50.18 |
| Tip-Adapter-F [21] | | 68.55 | 59.10 | 42.62 | 32.08 | 69.53 | 50.83 |
| HC-TR (Ours) | | 68.71 | 59.57 | 42.09 | 30.59 | 68.86 | 50.28 |
| HC-TAF (Ours) | | **69.04** | **59.75** | **42.74** | **32.16** | **69.60** | **51.06** |
| Zero-shot CLIP [1] | | 66.73 | 60.83 | 46.15 | 47.77 | 73.96 | 57.18 |
| Linear Probe CLIP [1] | | 65.85 | 56.26 | 34.77 | 35.68 | 58.43 | 46.29 |
| TaskRes [20] | ViT-B/16 | 73.55 | 65.81 | 48.86 | 49.85 | 77.35 | 60.47 |
| Tip-Adapter-F [21] | | 73.77 | 65.90 | 49.13 | **50.81** | 77.96 | 60.95 |
| HC-TR (Ours) | | 73.85 | 65.98 | 48.88 | 50.08 | 77.51 | 60.61 |
| HC-TAF (Ours) | | **74.08** | **66.18** | **49.30** | 50.75 | **78.01** | **61.06** |

**Domain Generalization.** We investigate the generalization ability of the models constrained by HC under domain shift. We train the models on ImageNet with 16 shots and test the trained models on ImageNet variant datasets ImageNet-V2, ImageNet-Sketch, ImageNet-A and ImageNet-R. As

shown in Tab. 2, HC-TR and HC-TAF outperform their respective baselines on the source dataset ImageNet across four different visual backbones ResNet-50, ResNet-101, ViT-B/32 and ViT-B/16. The performance of HC-TR is slightly inferior to that of TaskRes on ImageNet-Sketch, -A and -R with backbones ResNet-50, ResNet-101 and ViT-B/32, which concurs with TaskRes's reported source-overfitting pitfall. Nonetheless, HC-TR outperforms TaskRes with the more representative ViT-B/16. HC-TAF shows better generalization than Tip-Adapter-F in almost all target domains with various backbones. Experiments demonstrate that the performance improvement is not reliant on the shortcut to overfit the seen data domain.

### 4.3 Ablation Studies

**Constraint terms: track coincidence and deviating perturbation.** The homology consistency constraint consists of track coincidence and deviating perturbation, and the effect of homology consistency constraint in efficient transfer comes from their collaboration. Here, we investigate the individual roles of track coincidence and deviating perturbation in tuning. The ablation on ImageNet dataset is shown in Tab. 3, from which we can observe that (1) Without the generalization enhancement in latent spaces brought by DP, the performance

Table 3: The ablation studies for the constraint terms, track coincidence (TC) and deviating perturbation (DP).

| Baseline | TC | DP | 1-shot | 2-shot | 4-shot | 8-shot | 16-shot |
|---|---|---|---|---|---|---|---|
| | | | 61.44 | 62.28 | 62.78 | 64.30 | 65.41 |
| TaskRes | ✓ | | 61.46 | 62.38 | 62.95 | 64.40 | 65.74 |
| | | ✓ | - | 62.14 | 62.67 | 64.23 | 65.33 |
| | ✓ | ✓ | - | 62.34 | 62.86 | 64.47 | 65.90 |
| | | | 61.16 | 61.74 | 62.65 | 64.07 | 65.43 |
| Tip-Adapter-F | ✓ | | 61.17 | 61.76 | 62.70 | 64.22 | 65.99 |
| | | ✓ | - | 61.76 | 62.63 | 63.95 | 65.17 |
| | ✓ | ✓ | - | 61.79 | 62.73 | 64.37 | 66.04 |

of TC alone decreases relative to full homology consistency constraint. (2) Conversely, if we do not guide the coincidence of persistence tracks and only apply DP to track end-point samples, the original performance of baselines will be damaged. This is because DP plays a role of regularization term for TC. Without coinciding persistence tracks by TC, not only the direct constraint on the structural equivalence of latent manifolds is lost, but also the only DP will cause the track end-point samples to randomly deviate from the hetero-modal training samples, which interferes with the downstream tuning and thus performance drop.

**Scaling factors of HC gradients in tuning.** The hyper-parameters $\eta$ and $\omega$ scale the weight of homology consistency constraint in tuning and $\lambda$ controls the strength of deviating perturbation within the constraint (Sec. 3.3). We adopt the constant scaling (with $\eta$, $\lambda$) for TaskRes and adaptive scaling (with $\omega$, $\lambda$) for Tip-Adapter-F. In general, performance first increases and then decreases as $\eta$ (or $\omega$) and $\lambda$ increase, and the optimal performance is achieved by a specific combination of their values. Taking 16-shot ImageNet as an example, as shown in Tab. 4, the optimal $\eta$ and $\lambda$ are 15 (i.e., $0.15/\tau_{\text{CLIP}}$) and 2.5 on HC-TR, the optimal $\omega$ and $\lambda$ are 0.4 and

Table 4: The ablation studies for the scaling factors of gradients in homology consistency constraint.

| $\eta\,(\lambda = 2.5)$ | 5 | 10 | 15 | 20 | 25 |
|---|---|---|---|---|---|
| HC-TR | 65.67 | 65.83 | 65.90 | 65.79 | 65.72 |

| $\lambda\,(\eta = 15)$ | 1.5 | 2.0 | 2.5 | 3.0 | 3.5 |
|---|---|---|---|---|---|
| HC-TR | 65.83 | 65.88 | 65.90 | 65.87 | 65.82 |

| $\omega\,(\lambda = 100)$ | 0.1 | 0.2 | 0.3 | 0.4 | 0.5 |
|---|---|---|---|---|---|
| HC-TAF | 65.59 | 65.82 | 65.96 | 66.04 | 65.99 |

| $\lambda\,(\omega = 0.4)$ | 50 | 100 | 150 | 200 | 250 |
|---|---|---|---|---|---|
| HC-TAF | 66.03 | 66.04 | 66.00 | 66.00 | 65.98 |

100 on HC-TAF. In implementation, we first determine the values of scaling factors at 16-shot, and then migrate them to HC*-TR / HC*-TAF and other few-shot settings. The setting of hyper-parameter factors on other datasets follows a similar procedure.

## 5 Conclusions, Limitations and Future Work

In this work, we study the generalizability of image-text alignment adjusting in the efficient transfer of VLMs under a low-data regime. We propose to explicitly constrain the structural equivalence of image and text latent manifolds in downstream tuning and design a theoretically well-founded homology consistency constraint based on persistent homology for VLMs transfer. Our method constraint coincides the persistences of homology classes of topological features between image and text manifolds and applies a deviating perturbation to generalize the persistence coincidence to unseen data. Extensive experiments demonstrate the effectiveness and robustness of our method.

**Limitations and Future Work.** In structural equivalence constraint, we do not explore the effects of higher-dimensional homology classes in depth. Besides, following previous work on efficient transfer learning for VLMs, we only apply the proposed homology consistency constraint on a series of few-shot recognition tasks. As a next step, we will further extend the application scenarios of our method to other VLMs downstream tasks.

## Acknowledgments and Disclosure of Funding

This research is supported by Artificial Intelligence-National Science and Technology Major Project 2023ZD0121200, National Science Fund for Excellent Young Scholars No. 62222212 and National Natural Science Foundation of China No. 62336001.

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

# A  Simplicial Complex, Homology Class, Persistent Homology and Homotopy Equivalent Assumption

## A.1  Simplicial Complex

A simplicial complex consists of a set of simplices, such as points, line segments, triangles, and $p$-dimensional analogues. A simplex $\sigma$ with dimension $p$ is the convex hull of a set of $p + 1$ affinely independent points $(x_0, \ldots, x_p)$. For $0 \leq p' \leq p$, a $p'$-face of $\sigma$ is a $p'$-simplex that is the convex hull of a non-empty subset. A simplicial complex $K$ consists of a set of finite simplices that satisfy: $K$ contains every face of each simplex in $K$ and for any two simplices $\sigma, \tau \in K$, their intersection $\sigma \cap \tau$ is either empty or a face of both $\sigma$ and $\tau$. The dimension of $K$ is the maximum dimension of any simplex in $K$. Given a finite data point set $X$ in metric space $(M, d)$ and a real $a > 0$, the Vietoris-Rips complex $K_a^{\text{Rips}}(X)$ is the set of simplices $\sigma$ such that $d(x, x') \leq a$ for every pair of vertices of $\sigma$, and the Čech complex $K_a^{\text{Cech}}(X)$ is the set of simplices $\sigma$ such that the closed balls centering its vertices have a non-empty interaction.

## A.2  Homology Class

In complex $K$, let the number of $p$-simplices be $m_p$, a $p$-chain $c$ is a formal sum of $p$-simplices added with some coefficients, *i.e.*, $c = \sum_{i=1}^{m_p} \alpha_i \sigma_i$. In particular, the $p$-chains with $\mathbb{Z}_2$-additions form a chain group $C_p(K)$ where the identity is the chain $0 = \sum_{i=1}^{m_p} 0 \sigma_i$, and since $c + c = 0$, the inverse of a chain $c$ is itself. When $K$ is clear from the context, $C_p(K)$ can also be notated as $C_p$. Define a boundary operator $\partial_p$ on $p$-simplex $\sigma$ as a map that sends $\sigma$ to the $(p-1)$-chain consisting of $\sigma$'s $(p-1)$-faces referred as $\sigma$'s boundary, *i.e.*, $\partial_p \sigma = \sum_{i=0}^{p}(x_0, \ldots, \hat{x}_i, \ldots, x_p)$ where $\hat{x}_i$ indicates that $x_i$ is omitted. The boundary of a vertex is empty. The $\partial_p$ induces a homomorphism $\partial_p : C_p \to C_{p-1}$ that produces a $(p-1)$-chain when extended to a $p$-chain $c$ through $\partial_p c = \sum_{i=1}^{m_p} \alpha_i (\partial_p \sigma_i)$. Applying $\partial_p$ to the chain groups, a sequence of homomorphism, $0 = C_{k+1} \xrightarrow{\partial_{k+1}} C_k \xrightarrow{\partial_k} C_{k-1} \cdots C_1 \xrightarrow{\partial_1} C_0 \xrightarrow{\partial_0} C_{-1} = 0$, where $0$ denotes a trivial group, is obtained. A $p$-chain $c$ is a $p$-cycle if $c$ has empty boundary. All $p$-cycles together form a cycle group $Z_p$ under the $\mathbb{Z}_2$-addition operation inherited from $C_p$. $Z_p$ is the subgroup of $C_p$ that is mapped to the $0$ of $C_{p-1}$ by $\partial_p$, *i.e.*, $Z_p$ is the kernel of $\partial_p$. The image of the boundary operator $\partial_p$ is a subgroup of $(p-1)$-chains, called boundary group $B_{p-1} = \partial_p(C_p)$. For any $p$-simplex $\sigma$, every $(p-2)$-faces is involved in exactly two $(p-1)$-faces in $\partial_p \sigma$, thus $\partial_{p-1} B_{p-1} = 0$ for $p > 0$ and $B_{p-1} \subseteq Z_{p-1}$. The homology group $H_p$ is defined to classify the cycles in $Z_p$ by collecting those cycles that differ by a boundary into the same class. This is achieved by $H_p = Z_p / B_p$, taking the quotient of the $Z_p$ with boundary group $B_p$. By definition, the elements of $H_p$ are cosets of $B_p$ in $Z_p$, *i.e.*, $\{c + B_p \mid c \in Z_p\}$. For a cycle $c$, $c + B_p$ in $H_p$ forms its homology class $[c]$. Two cycles $c$ and $c'$ in the same class $[c] = [c']$ are homologous. In fact, $[c] = [c']$ holds if and only if $c \in c' + B_p$.

## A.3  Persistent Homology

For data point set $X$ in metric space $(M, d)$, define a function $f : M \to \mathbb{R}, f((x_0, \ldots, x_p)) = \max_{i,j \in \{0, \ldots, p\}} f((x_i, x_j))$ on simplices. Then, given a sequence of thresholds $a_1 \leq a_2 \leq \ldots, \leq a_n$, the growing sublevel sets $f^{-1}(-\infty, a]$ at these values give rise to a nested sequence of subcomplexes, $K_{a_1} \subseteq K_{a_2} \subseteq \cdots \subseteq K_{a_n}$, called a filtration $\mathcal{F}$. The inclusions in filtration $\mathcal{F}$ induce a sequence of homomorphisms $h_p^{i,j}$, $H_p \mathcal{F} : 0 = H_p(K_0) \to H_p(K_1) \to \cdots \to H_p(K_i) \xrightarrow{h_p^{i,j}} \cdots \to H_p(K_j) \to \cdots \to H_p(K_n) = H_p(K)$, called a homology module. The images of the homomorphisms $h_p^{i,j}$ are persistent homology groups $H_p^{i,j}$ for the module, $H_p^{i,j} = \text{im } h_p^{i,j}$, for $0 \leq i \leq j \leq n$. The non trivial elements of persistent homology groups $H_p^{i,j}$ consist of homology classes that survive from $K_i$ to $K_j$, *i.e.*, the homology classes that do not quotient out by boundaries, which implies that $H_p^{i,j} = Z_p(K_i) / (B_p(K_j) \cap Z_p(K_i))$. A non-trivial homology class $\epsilon \in H_p(K_a)$ is born at $K_i$, if $\epsilon \in H_p^{i,a}$ but $\epsilon \notin H_p^{i-1,a}$. Likewise, the $\epsilon$ dies entering $K_j$, if $\epsilon \in H_p^{a,j-1}$ but $\epsilon \notin H_p^{a,j}$. When a class dies, it may be merged with several classes and raises a new birth. The persistence of homology class that is created at $K_i$ and destoryed at $K_j$ is defined as $a_i - a_j$. The homology classes that never die are set to remain alive till $a_{n+1} = a_\infty = \infty$. As a visual representation, the persistence diagram

$\mathrm{Dgm}_p(\mathcal{F})$ of filtration $\mathcal{F}$ draws the paired birth at $a_i$ and death at $a_j$ that bound the survival interval of one or more homology classes as a point $(a_i, a_j)$ on the extended plane $\overline{\mathbb{R}}^2 := (\mathbb{R} \cup \{\pm\infty\})^2$.

### A.4 Assumption on Homotopy Equivalent of Image and Text Latent Manifolds

Topology specifies how points are connected and is a key geometric signature of metric spaces. Topological equivalence can be formalized by continuous functions that map points from one space to the other while preserving the connectivity. A straightforward equivalence is that the points as well as their neighborhoods in two spaces are in one-to-one correspondence, called being homeomorphic [24, 26], which requires a bijecitve map with continuous inverse. Two homeomorphic spaces share the exact same topological properties, *i.e.*, from the topological point of view, the two are completely consistent. Another less rigid relation is homotopy equivalent [24, 26]. Formally, given spaces $\mathbb{X}$ and $\mathbb{Y}$, two maps $f_0, f_1 : \mathbb{X} \to \mathbb{Y}$ are homotopic if they can be joined by a continuous function $F : \mathbb{X} \times [0,1] \to \mathbb{Y}$ such that $F(x,0) = f_0(x)$ and $F(x,1) = f_1(x)$ for every $x \in \mathbb{X}$, denoted as $f_0 \simeq f_1$. The spaces $\mathbb{X}$ and $\mathbb{Y}$ are homotopy equivalent if there exist two maps $f : \mathbb{X} \to \mathbb{Y}$ and $g : \mathbb{Y} \to \mathbb{X}$ such that $f \circ g \simeq \mathrm{id}_{\mathbb{Y}}$ and $g \circ f \simeq \mathrm{id}_{\mathbb{X}}$, which suggests an intuitive fact that space $\mathbb{X}$ is homotopy equivalent to $\mathbb{Y}$ if and only if both $\mathbb{X}$ and $\mathbb{Y}$ are deformation retracts of a common space $\mathbb{Z}$.

Vision and language are key information modalities for human cognition and have intrinsic correspondence on semantics. Observe that similar image scenes tend to be described by synonymic texts, that is, the connectivity of image data samples is preserved after mapping them to the text latent manifold, and vice versa. This preservation of connectivity suggests that the two manifolds have similar topological structures. Besides, image and text data are both relevant and complementary concrete expressions of the real world. The latent manifolds in which they reside can be seen as deformation retracts of the general world knowledge space. Thus, it is rational to consider that the image latent manifold and text latent manifold are homotogy equivalent. As a topological invariant of homotogy equivalent spaces, the homology is a viable tool for quantifying and aligning the structure of image and text latent manifolds.

## B  Full numerical results on few-shot learning

Table 5: Full numerical results of performance comparison on few-shot learning.

| Method | Setting | ImageNet | Caltech101 | DTD | EuroSAT | FGVCAircraft | Flowers102 | Food101 | OxfordPets | StanfordCars | SUN397 | UCF101 |
|---|---|---|---|---|---|---|---|---|---|---|---|---|
| Zero-Shot CLIP | | 58.18 | 86.29 | 42.32 | 37.56 | 17.28 | 66.14 | 77.31 | 85.77 | 55.61 | 58.52 | 61.46 |
| CLIP-Adapter | | 61.20 | 88.60 | 45.80 | 61.40 | 17.49 | 73.49 | 76.82 | 85.99 | 55.13 | 61.30 | 62.20 |
| TaskRes | | 61.44 | 89.49 | 49.65 | 63.52 | 21.36 | 79.09 | 75.03 | 84.60 | 59.78 | 61.87 | 64.26 |
| Tip-Adapter-F | | 61.16 | 89.21 | 49.17 | 59.10 | 20.46 | 79.29 | 77.52 | 86.43 | 59.69 | 61.95 | 65.05 |
| HC-TR | 1-shot | 61.46 | 89.78 | 49.88 | 63.43 | 21.39 | 79.37 | 75.25 | 84.66 | 59.68 | 62.13 | 64.16 |
| HC*-TR | | 62.15 | 89.61 | 49.70 | 64.02 | 21.48 | 80.02 | 75.47 | 84.57 | 59.84 | 62.92 | 64.79 |
| HC-TAF | | 61.17 | 89.57 | 50.00 | 61.70 | 20.67 | 80.23 | 77.59 | 86.64 | 59.61 | 62.03 | 65.69 |
| HC*-TAF | | 62.04 | 89.33 | 50.77 | 60.65 | 21.39 | 80.39 | 77.89 | 86.84 | 59.56 | 63.75 | 66.51 |
| Zero-Shot CLIP | | 58.18 | 86.29 | 42.32 | 37.56 | 17.28 | 66.14 | 77.31 | 85.77 | 55.61 | 58.52 | 61.46 |
| CLIP-Adapter | | 61.52 | 89.37 | 51.48 | 63.90 | 20.10 | 81.61 | 77.22 | 86.73 | 58.74 | 63.29 | 67.12 |
| TaskRes | | 62.28 | 89.86 | 54.43 | 65.99 | 23.67 | 83.09 | 75.88 | 85.64 | 62.70 | 64.89 | 67.41 |
| Tip-Adapter-F | | 61.74 | 89.86 | 54.37 | 66.19 | 23.34 | 85.30 | 77.73 | 87.05 | 61.52 | 64.25 | 68.28 |
| HC-TR | 2-shot | 62.34 | 89.90 | 54.79 | 66.91 | 24.63 | 83.11 | 76.03 | 85.66 | 63.06 | 65.04 | 67.57 |
| HC*-TR | | 62.59 | 90.14 | 55.67 | 66.22 | 25.20 | 85.71 | 75.81 | 85.83 | 64.48 | 65.41 | 69.18 |
| HC-TAF | | 61.79 | 90.26 | 54.61 | 67.53 | 24.21 | 85.99 | 77.78 | 87.22 | 63.11 | 64.42 | 68.31 |
| HC*-TAF | | 62.61 | 90.63 | 56.09 | 67.33 | 24.69 | 87.49 | 77.65 | 88.14 | 64.48 | 65.77 | 69.94 |
| Zero-Shot CLIP | | 58.18 | 86.29 | 42.32 | 37.56 | 17.28 | 66.14 | 77.31 | 85.77 | 55.61 | 58.52 | 61.46 |
| CLIP-Adapter | | 61.84 | 89.98 | 56.86 | 73.38 | 22.59 | 87.17 | 77.92 | 87.46 | 62.45 | 65.96 | 69.05 |
| TaskRes | | 62.78 | 90.71 | 59.40 | 74.42 | 26.49 | 89.97 | 76.45 | 87.03 | 66.60 | 67.22 | 69.36 |
| Tip-Adapter-F | | 62.65 | 90.95 | 58.22 | 73.46 | 25.98 | 89.69 | 78.12 | 87.49 | 65.73 | 67.13 | 71.05 |
| HC-TR | 4-shot | 62.86 | 90.83 | 59.40 | 75.02 | 26.76 | 90.86 | 76.49 | 87.10 | 66.98 | 67.34 | 69.92 |
| HC*-TR | | 63.60 | 91.60 | 59.34 | 75.14 | 27.69 | 92.13 | 76.86 | 88.54 | 68.54 | 67.36 | 71.64 |
| HC-TAF | | 62.73 | 91.44 | 58.87 | 78.54 | 28.65 | 90.99 | 78.20 | 87.71 | 67.07 | 67.06 | 72.03 |
| HC*-TAF | | 63.28 | 91.76 | 59.99 | 78.74 | 28.80 | 91.80 | 78.53 | 88.17 | 68.29 | 67.84 | 73.17 |
| Zero-Shot CLIP | | 58.18 | 86.29 | 42.32 | 37.56 | 17.28 | 66.14 | 77.31 | 85.77 | 55.61 | 58.52 | 61.46 |
| CLIP-Adapter | | 62.68 | 91.40 | 61.00 | 77.93 | 26.25 | 91.72 | 78.04 | 87.65 | 67.89 | 67.50 | 73.30 |
| TaskRes | | 64.30 | 92.09 | 63.48 | 78.07 | 32.25 | 94.05 | 77.82 | 87.35 | 70.70 | 69.29 | 74.46 |
| Tip-Adapter-F | | 64.07 | 92.25 | 63.12 | 78.04 | 32.16 | 91.76 | 78.52 | 87.79 | 70.12 | 69.43 | 74.97 |
| HC-TR | 8-shot | 64.47 | 92.41 | 63.36 | 78.35 | 32.34 | 94.19 | 77.98 | 87.79 | 71.38 | 69.37 | 74.81 |
| HC*-TR | | 64.81 | 92.62 | 64.30 | 78.96 | 33.57 | 94.88 | 77.99 | 88.12 | 73.03 | 69.39 | 75.84 |
| HC-TAF | | 64.37 | 92.45 | 63.83 | 81.35 | 34.62 | 93.02 | 78.92 | 88.61 | 71.98 | 69.74 | 76.02 |
| HC*-TAF | | 64.86 | 93.06 | 65.13 | 81.83 | 35.52 | 94.36 | 78.86 | 89.13 | 73.39 | 69.88 | 77.24 |
| Zero-Shot CLIP | | 58.18 | 86.29 | 42.32 | 37.56 | 17.28 | 66.14 | 77.31 | 85.77 | 55.61 | 58.52 | 61.46 |
| CLIP-Adapter | | 63.59 | 92.49 | 65.96 | 84.43 | 32.10 | 93.9 | 78.25 | 87.84 | 74.01 | 69.55 | 76.76 |
| TaskRes | | 65.41 | 93.31 | 67.02 | 83.25 | 35.82 | 95.53 | 78.33 | 88.53 | 77.07 | 71.41 | 78.43 |
| Tip-Adapter-F | | 65.43 | 93.06 | 66.55 | 84.98 | 36.42 | 94.64 | 79.4 | 89.67 | 75.80 | 71.52 | 78.09 |
| HC-TR | 16-shot | 65.90 | 93.59 | 67.61 | 83.19 | 37.20 | 96.06 | 78.57 | 88.93 | 77.61 | 71.64 | 78.54 |
| HC*-TR | | 66.25 | 93.75 | 67.85 | 83.78 | 38.25 | 95.98 | 78.49 | 88.96 | 77.75 | 71.34 | 78.20 |
| HC-TAF | | 66.04 | 93.35 | 68.32 | 86.65 | 38.34 | 95.37 | 79.54 | 90.49 | 77.55 | 72.01 | 79.22 |
| HC*-TAF | | 66.40 | 93.83 | 68.56 | 85.73 | 38.82 | 95.82 | 79.61 | 90.24 | 77.64 | 71.62 | 79.54 |

