# OpenReview forum: "Homology Consistency Constrained Efficient Tuning for Vision-Language Models"
_NeurIPS.cc/2024/Conference — NeurIPS 2024 poster_

### Official Review · Reviewer_zhsp · 2024-07-12

**Soundness:** 2
**Presentation:** 2
**Contribution:** 2
**Rating:** 5
**Confidence:** 3

**Summary:**

A Homology Consistency (HC) constraint for efficient transfer on VLMs is proposed in this paper, which explicitly constrains the correspondence of image and text latent manifolds through structural equivalence based on persistent homology in downstream tuning.
The proposed method tracks the persistence of the homology classes of topological features across multiple scales and guide the directions of persistence tracks in image and text manifolds to coincide each other. Additionally, a deviating perturbation is applied to generalize the persistence coincidence to unseen data. Experiments on recognition and generalization tasks show the superior performance.

**Strengths:**

1. The paper is well-written with a straightforward motivation.
2. A theoretically well-founded homology consistency (HC) constraint based on persistent homology is proposed for efficient transfer on VLMs.
3. Experiments on recognition tasks show the superior performance.

**Weaknesses:**

The hyper-parameters η, λ, ω should be determined at 16 shots and then migrated to other few-shot settings. If the number of samples is less than 16, how should the aforementioned hyper-parameters be set, and will there be a significant difference in performance?

**Questions:**

The hyper-parameters η, λ, ω should be determined at 16 shots and then migrated to other few-shot settings. If the number of samples is less than 16, how should the aforementioned hyper-parameters be set, and will there be a significant difference in performance?

**Limitations:**

The experiments only conducted on recognition tasks.

---

> ### Author Rebuttal · Authors · 2024-08-07
>
> Thank you for your valuable concerns.
>
> **W1&Q1**: Referring to the widely-acknowledged evaluation standard in the same field, we conduct experiments on few-shot learning in the 1-/2-/4-/8-/16-shot setting. We find that perturbations near the optimal scaling hyper-parameters $\eta$, $\lambda$, $\omega$ have little effect on model performance, as shown in the ablation study of hyper-parameters $\eta$, $\lambda$, $\omega$ in our paper. That is, our performance is robust to these hyper-parameters. Therefore, we determine the values of these hyper-parameters in the setting with the largest number of shots (i.e., 16) and transfer them to other shot settings on the same dataset.
>
> **L1**: We would like to clarify why we conducted on  few-shot recognition tasks.
>
> The efficient transfer learning on VLMs that we focus on is to tune the large-scale VLMs toward downstream tasks under low-data regime. It aims to achieve considerable improvements in tuning VLMs on target tasks with limited data resources.
>
> Few-shot recognition is a *widely-acknowledged evaluation standard* for the efficient transfer learning. Specifically, based on limited samples per class, few-shot  recognition on VLMs is to tune the pre-trained semantic alignment to handle the correspondence between images and category texts in target domain. Therefore, the performance on few-shot recognition can effectively reflect the ability of efficient transfer.
>
> In our paper, we conduct experiments on few-shot recognition over 11 datasets (covering a wide range of visual recognition on generic objects, fine-grained categories, scenes, actions, etc.), domain generalization over 4 visual backbones from ImageNet source to 4 target domains ImageNet-V2/-Sketch/-A/-R, and a series of necessary ablation studies, to demonstrate the effectiveness and robustness of our method.

---

### Official Review · Reviewer_KyVk · 2024-07-13

**Soundness:** 3
**Presentation:** 3
**Contribution:** 2
**Rating:** 5
**Confidence:** 3

**Summary:**

The paper identifies a key issue with existing methods for tuning pre-trained vision-language models to downstream tasks with limited data: they adjust the alignment between image and text based solely on observed samples, which may not generalize well beyond the training data. To address this issue, the paper proposes a novel constraint from the perspective of topological data analysis.

This constraint employs persistent homology to ensure the structural equivalence of image and text latent manifolds during tuning.

**Strengths:**

1.  The paper offers a new way of looking at model tuning through the lens of topological analysis, with a focus on understanding the structure of data spaces for better semantic alignment in vision-language tasks. I appreciate this perspective on the issue.

2.  The proposed method exhibits a thoughtful theoretical underpinning, using persistent homology to enhance the generalizability of image-text alignment adjusting.&#x20;

3.  The paper is well-written and the reason for leveraging topological data analysis to enhance semantic alignment during the tuning process is reasonable and easy to follow up.

**Weaknesses:**

The paper does not adequately discuss how it relates to existing image and text alignment techniques, including those based on distance metrics, mutual information, adversarial training, and attention mechanisms. This lack of comparative analysis creates a gap in fully appreciating the distinctive contributions and potential advantages.

**Questions:**

1.  Could you please provide insights into the fundamental differences and advantages of topological data analysis in your method over other alignment methods?

2.  Where do you foresee potential challenges in extending your proposed method to tasks outside the few-shot learning domain, particularly in scenarios such as zero-shot learning, or applications involving detection and segmentation?

3.  Could you elaborate on how incorporating higher-dimensional homology classes into the tuning process might impact the model's performance and behavior, beyond the computational cost?

**Limitations:**

Yes

---

> ### Author Rebuttal · Authors · 2024-08-07
>
> Thank you for your constructive comments and insightful questions.
>
> **W1&Q1**: Our main differences and advantages over existing image-text alignment techniques are that we explicitly constrain the structural equivalence between image and text latent manifolds, and achieve a topological consistency of the latent manifolds beyond the localized closeness of discrete samples, so as to enhance the generalization of the learned image-text alignment.
>
> In the widely-used alignment techniques, contrastive learning (e.g., InfoNCE loss, triplet loss) forces semantically related image-text pairs closer and pull unrelated ones away, to enhance the discrimination in the common embedding space. Mutual information maximization is to increase the correspondence probability between semantically related samples to overlap image and text latent manifolds for aligning. Cross-modal attention mechanism learns to focus on similar content in images and texts, thereby capturing the semantic associations across modalities accurately.
>
> Despite various approaches, the focus of existing alignment techniques tends to be limited to the localized closeness on the observed data samples, due to the lack of perspective on the underlying structure of image and text latent manifolds in training. The learned alignment cannot guarantee its generalization beyond training samples, especially in low-data regime.
>
> Our theoretically well-founded HC constructs the discrete analogues of image and text latent manifolds based on data samples, explicitly constrain the structural equivalence between image and text latent manifolds, and achieve a topological consistency on the underlying manifold structures beyond the localized closeness of discrete samples. As a constraint term in the efficient transfer learning on VLMs with limited data resources in our paper, the effectiveness and robustness of HC are demonstrated by extensive experiments.
>
> **Q2**: Thank you for your constructive comments.
>
> For zero-shot learning: In capturing the structural association between a given test sample of unseen category and the training samples in their latent manifold for zero-shot inference, The embedding of test samples of unseen categories may introduce noise into the construction of simplicial complex and topological feature extraction. In addition, if the volume of pre-training set is too large, it is costly to obtain topological structure around the given test sample in manifold.
>
> For image understanding tasks such as detection and segmentation: Unlike manifolds in semantic space considered by our HC, in the pixel space of interest for object detection and segmentation, the topological information of objects, such as shape, connectivity, and boundaries, is more intricate, thus requiring more sensitive topological structure representing methods. Meanwhile, topology information data in these fields is scarce, and annotation requires specialized knowledge.
>
> **Q3**: Beyond the computational cost, we find that adding 1-st homology consistency in addition to 0-th homology consistency brings almost no additional performance improvement in tuning VLMs. We conjecture it is because the 1-st homology classes (i.e., loops) are too intricate for constraining the structural equivalence of image and text latent manifolds, and the existing VLMs' ability to represent data samples is insufficient to support the alignment of such precise structures. In addition, the limited data samples may not be sufficient for capturing the persistence of higher-dimensional homology features in latent manifolds.

---

### Official Review · Reviewer_Ls33 · 2024-07-14

**Soundness:** 3
**Presentation:** 3
**Contribution:** 2
**Rating:** 6
**Confidence:** 4

**Summary:**

The paper introduces a Homology Consistency (HC) constraint for efficient transfer learning on vision-language models (VLMs), ensuring task-specific image-text alignment while preserving general knowledge by using structural equivalence based on persistent homology. This approach mimics the topology of latent manifolds and tracks the persistence of topological features.

**Strengths:**

1. This paper is well motivated, and the motivation of using homology consistency is interesting.
2. This paper has a good theoretic support.

**Weaknesses:**

1. The performance of the proposed method is worse than the baseline method in low-shot (1-shot and 2-shot) tasks.
2. The improvement in Table 2 is marginal. Is the comparison fair with the same random seed? How many runs did you conduct? Could the authors also report the standard deviation of the score?
3. Moreover, is 16-shot common in this benchmark? 16 shot seems a lot in few-shot learning.
4. Can you also elaborate more why with only DP, the performance drops in Table 3?
5. In addition, could you elaborate more why choosing 0-th homology classes? What are the potential effects of using other homology classes?

**Questions:**

1. Can you visualise the topology of the data before and after adaptation with the HC constraint? It will be very interesting to see how actually the HC constraint can preserve the topology of the manifold during transfer learning.

**Limitations:**

My concern is the performance improvement is marginal and limited to more shots setting (16-shot).

---

> ### Author Rebuttal · Authors · 2024-08-07
>
> Thank you for your constructive comments.
>
> **W1**: Our main results, the performance comparisons between baselines and our proposed HC/HC* on 1-/2-/4-/8-/16-shot settings over 11 benchmark datasets are shown in Fig. 3, and the corresponding detailed numerical results are in Appendix C. It can be found that HC/HC*+TaskRes and HC/HC*+Tip-Adapter-F outperform their baselines on 1-shot and 2-shot settings in most cases. We wonder if we misinterpreted your question due to some misunderstanding. We sincerely look forward to your reply.
>
> **W2**: Thank you for your concern. Experiments on baseline and with HC / HC* constraint are conducted in the same setting (as stated in Sec. 4.1 Experimental settings). For all experiments reported in paper, we fixed the random seed as 1. Our performance comparison is fair.
>
> Here, to verify the robustness of our improvement, we take 16-shot ImageNet as an example, and set the random seed from 1 to 4. The test accuracies of HC*+Tip-Adapter-F, HC*+TaskRes and their baselines are listed below.
>
> |      Method      | random seed 1 | random seed 2 | random seed 3 | random seed 4 |
> |:----------------:|:-------------:|:-------------:|:-------------:|:-------------:|
> |      TaskRes     |     65.41%    |     65.47%    |     65.49%    |     65.47%    |
> |    HC*+TaskRes    |     66.25%    |     66.13%    |     66.19%    |     66.24%    |
> |   Tip-Adapter-F  |     65.43%    |     65.05%    |     65.15%    |     64.99%    |
> | HC*+Tip-Adapter-F |     66.40%    |     66.27%    |     66.14%    |     66.05%    |
>
> As shown in table, the average gain of HC*+Tip-Adapter-F compared to Tip-Adapter-F is 1.06% and the standard deviation is 0.098%. The average gain of HC*+TaskRes compared to TaskRes is 0.74% and the standard deviation is only 0.069%.
>
> In fact, our performance improvements are very competitive in the field of efficient transfer learning of VLMs. On the same evaluation on 16-shot ImageNet, GraphAdapter [1] (NeurIPS 2023) improves by 0.95% over the SOTA it compares to (refer to Tab. 1 in GraphAdapter [1]), APE-T  [2] (ICCV 2023) outperforms its baseline Tip-Adapter-F by 0.56% (refer to the numerical results APE-T [2] released), which is significantly lower than our average gain of 1.06% on the same baseline.
>
> [1] Graphadapter: Tuning vision-language models with dual knowledge graph.
>
> [2] Not all features matter: Enhancing few-shot clip with adaptive prior refinement.
>
> **W3**: Following the commonly-used evaluation standard in previous work, we perform experiments on few-shot learning in 1-/2-/4-/8-/16-shot settings over 11 datasets (Fig. 3 in paper), and domain generalization on 16-shot ImageNet across 4 visual backbones (Tab. 2 in paper). Meanwhile, in transferring pre-trained VLMs, compared to the web-scale volume of pre-training data, the tuning data of 16 samples per class is significantly less, so it is reasonable to evaluate efficient transfer learning on VLMs in 16-shot setting.
>
> **W4**: Our proposed homology consistency (HC) constraint consists of track coincidence (TC) term and deviating perturbation (DP) term. TC term is responsible for guiding the persistence tracks in image and text manifolds to coincide each other to achieve a homology-level structural equivalence, and DP term drives end-point samples of tracks to deviate from their semantically related hetero-modal seen samples toward unseen ones in embedding (to be distributed uniformly), to enhance the generalization of TC in constraining structural equivalence of latent manifolds.
>
> In other words, TC is leading role in achieving homology consistency, and DP can be regarded as its regularization term. Without coinciding persistence tracks by TC, not only the direct constraint on the structural equivalence of latent manifolds is lost, but also the only DP will cause the track end-point samples to randomly deviate from the hetero-modal training samples, which interferes with the downstream tuning and thus performance drop.
>
> **W5**: We mainly consider two aspects: computational cost and efficacy. *First*, in practice, the training time for constraining both 0-th and 1-st persistence tracks to coincide is about 20 times that for only the 0-th track coincidence. *Secondly*, we find that in tuning VLMs, adding 1-st homology consistency in addition to 0-th homology consistency brings almost no additional gain. We conjecture it is because the 1-st homology classes (i.e., loops) are too intricate for constraining the structural equivalence of image and text latent manifolds, and the existing VLMs' ability to represent data samples is insufficient to support the alignment of such precise structures.
>
> **Q1**: Following your suggestion, we visualize the persistence intervals of homology classes from their birth to death time through persistence diagram (PD), to show the change in the topology of data between the tuning with and without our HC constraint. Persistence diagram draws the paired birth at $b$ and death at $d$ that bound the survival interval of homology classes as a point $(b, d)$, which is a widely-used visual representation of persistent homology in topological data analysis.
>
> Specifically, we visualize the persistence of homology classes of Rips complexes built on the top of test images, text embeddings tuned by TaskRes without HC constraint and text embeddings tuned by HC+TaskRes after applying HC, respectively, on 4 datasets including 2 generic object datasets, ImageNet and Caltech101, and 2 fine-grained object datasets, FGVCAircraft and StanfordCars.
>
> As shown in Fig. 1 in the PDF we submitted, where the coordinates of the red dots represent the paired birth time and death time of the corresponding 0-th homology classes (Note that 0-th homology classes all emerge at 0). It can be found that, compared with the text manifold tuned without HC, the 0-th homology persistence of text manifold tuned by HC+TaskRes is evidently more akin to the image latent manifold.

---

### Official Review · Reviewer_yCSE · 2024-07-17

**Soundness:** 2
**Presentation:** 2
**Contribution:** 2
**Rating:** 3
**Confidence:** 4

**Summary:**

This paper proposes Homology Consistency (HC) constraint for transfer learning on VLMs, and it explicitly constrains the correspondence of image and text latent manifolds by structural equivalence based on persistent homology in downstream tuning.

**Strengths:**

1. The proposed method is well-founded and clearly explains the proposed homology consistency (HC) constraint.

2. Extensive experiments are performed on 11 benchmark datasets.

**Weaknesses:**

1. The paper lacks discussions on the computational cost of the proposed techniques.

2. The proposed method for constraining the structural equivalence of image and text latent manifolds seems generalizable to other learning tasks for vision-language models. However, the proposed method is only evaluated for few-shot learning of vision language models.

3. Although the model outperforms other methods in most cases, the improvements are relatively marginal.

4. The paper only applies the method to a limited number of adapter models (TaskRes and Tip-Adapter-F).

**Questions:**

see weaknesses.

**Limitations:**

The limitations are not discussed sufficiently (see weaknesses).

---

> ### Author Rebuttal · Authors · 2024-08-07
>
> Thank you for your valuable concerns, and we would like to clarify as follows.
>
> **W1**: Since our proposed HC constraint incurs no additional cost in inference and the cost increase in offline training is marginal (less then 1.0%), we do not discuss computational cost in our paper. Following your suggestion, we take the transfer of VLMs on 16-shot ImageNet as an example to analyze the training cost of our HC. The training time of baseline Tip-Adapter-F is 15 s/epoch, and that of HC+Tip-Adapter-F constrained by our HC is 21 s/epoch. The peak total GPU memory usage of baseline Tip-Adapter-F is 16707MB, and 16751MB for HC+Tip-Adapter-F, only a 0.26% increase. We will add a discussion on computational cost in revision.
>
> **W2**: The efficient transfer learning on VLMs is to tune the large-scale VLMs (e.g., CLIP) toward downstream tasks under low-data regime. It aims to achieve considerable improvements in tuning VLMs on target tasks with limited data resources.
>
> Few-shot learning is a *widely-acknowledged evaluation standard for the efficient transfer learning*. Specifically, based on limited samples per class, few-shot learning on VLMs is to tune the pre-trained semantic alignment to handle the correspondence between images and category texts in target domain. Therefore, the performance on few-shot learning can effectively reflect the ability of efficient transfer.
>
> In our paper, we conduct experiments on few-shot learning over 11 datasets (covering a wide range of visual recognition on generic objects, fine-grained categories, scenes, actions, etc.), domain generalization over 4 visual backbones from ImageNet source to 4 target domains ImageNet-V2/-Sketch/-A/-R, and a series of necessary ablation studies, to demonstrate the effectiveness and robustness of our method.
>
> **W3**: The performance improvements brought by our method are robust and reliable. For instance, we fix the random seed in training on 16-shot ImageNet from 1 to 4, and the performance of the baselines and constrained by our method are shown below.
>
> |      Method      | random seed 1 | random seed 2 | random seed 3 | random seed 4 |
> |:----------------:|:-------------:|:-------------:|:-------------:|:-------------:|
> |      TaskRes     |     65.41%    |     65.47%    |     65.49%    |     65.47%    |
> |    HC*+TaskRes    |     66.25%    |     66.13%    |     66.19%    |     66.24%    |
> |   Tip-Adapter-F  |     65.43%    |     65.05%    |     65.15%    |     64.99%    |
> | HC*+Tip-Adapter-F |     66.40%    |     66.27%    |     66.14%    |     66.05%    |
>
> As shown in table, the average performance gain of HC*+Tip-Adapter-F over baseline Tip-Adapter-F is 1.06%, and the standard deviation is 0.098%. The average gain of HC*+TaskRes over TaskRes is 0.74%, and the standard deviation is only 0.069%.
>
> In fact, our performance improvements are very competitive in the field of efficient transfer learning of VLMs. On the same evaluation on 16-shot ImageNet, GraphAdapter [1] (NeurIPS 2023) improves by 0.95% over the SOTA it compares to (refer to Tab. 1 in GraphAdapter [1]), APE-T  [2] (ICCV 2023) outperforms its baseline Tip-Adapter-F by 0.56% (refer to the numerical results APE-T [2] released), which is significantly lower than our average gain of 1.06% on the same baseline.
>
> [1] Li X, Lian D, Lu Z, et al. Graphadapter: Tuning vision-language models with dual knowledge graph[J]. Advances in Neural Information Processing Systems, 2023, 36.
>
> [2] Zhu X, Zhang R, He B, et al. Not all features matter: Enhancing few-shot clip with adaptive prior refinement[C]//Proceedings of the IEEE/CVF International Conference on Computer Vision. 2023: 2605-2615.
>
> **W4**: We would like to clarify why we take TaskRes and Tip-Adapter-F as the baselines for applying our proposed HC constraint. TaskRes and Tip-Adapter-F belong to the two mainstream paradigms of adapter tuning (residual blending based and key-value cache based), respectively. These two paradigms cover almost all existing adapter tuning methods of efficient transfer on VLMs. In our paper (Tab. 1), we compare our performance with a range of SOTA adapter methods, e.g., residual blending based CLIP-Adapter, TaskRes, GraphAdapter, and key-value cache based Tip-Adapter-F, APE-T, etc, and outperform them on the most evaluation.
>
> The models TaskRes and Tip-Adapter-F are representative in their respective paradigms. With the common residual blending idea in the same paradigm, the residual setting of TaskRes is relatively simple yet effective. For the key-value cache based paradigm, Tip-Adapter-F is the pioneering work that established the cache framework and subsequent methods such as APE-T are also improved from it.
>
> For the efficient transfer learning on VLMs, our HC constraint proposes to preserve the vision-language alignment of pre-trained general concepts by constrain the topological equivalence between image and text latent manifolds in downstream tuning, to enhance the generalization of tuned alignment. This perspective has not been explored by existing VLMs transfer methods, and is verified to be effective in extensive experiments on two representative baselines. We believe that our HC constraint can be extended to other efficient tuning methods.

---

### Author Rebuttal · Authors · 2024-08-07

We sincerely thank all reviewers for reviewing this paper, and have provided detailed responses to all the concerns raised by the reviewers.

---

> ### Author Response · Authors · 2024-08-12
>
> Dear reviewers:
>
> Thank you for the comments on our paper. We are grateful for your recognition of our motivation and technical contributions. Specifically, the reviewers encouraged that our method is theoretically well-founded (Reviewers yCSE, KyVk and zhsp), well-motivated (Reviewers Ls33) and has interesting perspective (Reviewer KyVk), our paper has a good theoretic support (Reviewer Ls33), is well-written (Reviewers KyVk and zhsp) and easy to follow-up (Reviewer KyVk).
>
> We have provided detailed responses to all concerns in the rebuttal. We find that the more common concerns lie in the evaluation approach (Reviewers yCSE and zhsp) and performance improvement (Reviewers yCSE and Ls33). In our rebuttal, we clarify in detail that our evaluation scheme is the widely-acknowledged standard in the field of VLMs efficient transfer learning and commonly-used in previous work, and our performance improvement is relatively very competitive in the same field.
>
> We appreciate all valuable comments. Please let us know if you have additional questions so that we can address them during the discussion period. If our responses address your concerns, we hope that you can kindly consider updating the score.
>
> Thank you all again.

---

### Comment · Area_Chair_3au9 · 2024-08-10
**Please address rebuttal**

Dear reviewers,

Thank you for your reviews. The authors have tried to address your concerns in the rebuttal. If you have not already done so, please carefully read their rebuttal, and let them know what you think, and whether there is any more clarifications you require. Note that author-reviewer discussion ends on August 13th.
Thanks!
the AC

---

### Author Response · Authors · 2024-08-13

Dear reviewers:

Thank you for the comments on our paper. We are grateful for your recognition of our motivation and technical contributions, and have provided detailed responses to all concerns in the rebuttal. We would appreciate it if you kindly let us know of any other concerns you may have and if we can be of any further assistance in clarifying them.  If our responses address your concerns, we hope that you can kindly consider updating the score.

Thank you once again for your contribution to the development of our paper.

---

### Decision · Program_Chairs · 2024-09-25

**Decision:**

Accept (poster)

**Comment:**

This paper suggests a novel homology consistency constraint to align language and vision representations for VLM transfer learning. Adding these constraints to two baseline transfer learning methods (TaskRes and Tip-Aligner-F) yields consistent improvements.  The reviewers agreed this approach has potential, but voiced some concerns regarding the experiments, mainly that they are only for few-shot classification, and that in some scenarios the results are not state of the art. Nonetheless, I feel this paper should be accepted: (a) because of the original idea of using persistence homology for VLMs and (b) because the topological regularization does consistently improve upon the backbone methods